# Inhibitory Effect of pH-Responsive Nanogel Encapsulating Ginsenoside CK against Lung Cancer

**DOI:** 10.3390/polym13111784

**Published:** 2021-05-28

**Authors:** Ziyang Xue, Rongzhan Fu, Zhiguang Duan, Lei Chi, Chenhui Zhu, Daidi Fan

**Affiliations:** 1Shaanxi Key Laboratory of Degradable Biomedical Materials and Shaanxi R&D Center of Biomaterials and Fermentation Engineering, School of Chemical Engineering, Northwest University, Taibai North Road 229, Xi’an 710069, China; jjjamiexzy@163.com (Z.X.); rongzhanfu@nwu.edu.cn (R.F.); duanzhiguang@nwu.edu.cn (Z.D.); zhuzhu@163.com (L.C.); 2Biotech & Biomed Research Institute, Northwest University, Taibai North Road 229, Xi’an 710069, China

**Keywords:** ginsenoside CK, nanogel, pH responsive, antitumor

## Abstract

Ginsenoside CK is one of the intestinal bacterial metabolites of ginsenoside prototype saponins, such as ginsenoside Rb1, Rb2, Rc, and Rd. Poor water solubility and low bioavailability have limited its application. The nanogel carriers could specifically deliver hydrophobic drugs to cancer cells. Therefore, in this study, a nanogel was constructed by the formation of Schiff base bonds between hydrazide-modified carboxymethyl cellulose (CMC-NH_2_) and aldehyde-modified β-cyclodextrin (β-CD-CHO). A water-in-oil reverse microemulsion method was utilized to encapsulate ginsenoside CK via the hydrophobic cavity of β-CD. β-CD-CHO with a unique hydrophobic cavity carried out efficient encapsulation of CK, and the drug loading and encapsulation efficiency were 16.4% and 70.9%, respectively. The drug release of CK-loaded nanogels (CK-Ngs) in vitro was investigated in different pH environments, and the results showed that the cumulative release rate at pH 5.8 was 85.5% after 140 h. The methylthiazolyldiphenyl-tetrazolium bromide (MTT) toxicity analysis indicated that the survival rates of A549 cells in CK-Ngs at 96 h was 2.98% compared to that of CK (11.34%). In vivo animal experiments exhibited that the inhibitory rates of CK-Ngs against tumor volume was 73.8%, which was higher than that of CK (66.1%). Collectively, the pH-responsive nanogel prepared herein could be considered as a potential nanocarrier for CK to improve its antitumor effects against lung cancer.

## 1. Introduction

In recent years, hydrophobic ginsenosides with broad anticancer activity had been considered as one of the most promising Chinese herbal medicines. They were divided into protopanaxadiol (PPDs) and protopanaxatriol (PPTs), which demonstrate a variety of pharmacological properties [1]. Ginsenoside compound K (CK) is an initial bacterial metabolite of PPD-type ginsenoside compounds. It has a wide range of pharmacological properties, such as anti-cancer, anti-inflammatory, anti-aging, anti-allergenic, and anti-diabetic [2,3,4]. Despite many advantages, low water solubility and low permeability greatly reduce the efficacy of CK and limit its clinical application [5].

β-cyclodextrin (β-CD), consisting of seven glucose units, possesses a truncated cone structure containing a hydrophobic inner ring and a hydrophilic outer ring [6,7]. It can form inclusion complexes with many drugs through hydrophobic, electrostatic, and van der Waals interactions, or hydrogen bonds [8,9,10]. In recent years, increasingly β-CD hydrogels have been developed for drug sustained release. The integration of β-cyclodextrin into a cross-linked polymer network improved the ability of hydrogels to deliver drugs and control drug release [11].

Nowadays, nanogels had received great attention as an excellent nano-drug delivery system [12], due to their unique characteristics, including sustaining drug release and adjustable particle size [13]. In addition, proper modification of nanogels could increase the encapsulation efficiency of insoluble anticancer drugs. In order to allow more precise release of drugs around tumors, nanogels that respond to various environment stimuli, such as temperature, magnetic fields and pH had been developed. These response modes caused the instability of nanogels in special forms, thereby increasing the selectivity and efficiency of drug delivery [14,15,16]. The pH of the tumor microenvironment is low (5.8–6.5) due to an augmented glycolysis and hypoxia typical of tumor cells [17,18]. Therefore, the pH-responsive type is an important feature of the controlled release of the drug at the target [19].

Therefore, the purpose of this study was to prepare a new type of pH-responsive nanogel, which could encapsulate the hydrophobic drug CK, control its release, and enhance its anti-cancer activity. The main synthesis process is shown in Figure 1. Hydrazide-modified CMC and aldehyde-modified β-CD acted as the main materials for the nanogel synthesis. CK was encapsulated in the unique hydrophobic cavity of β-CD, which greatly increased the CK loading up to 16.4%. In addition, the Schiff base structure of the resulting nanogels formed by the hydrazide group and the aldehyde group was sensitive to pH and could be broken in an acidic environment. Hence, effective intracellular drug release could be achieved in the acidic tumor microenvironment and enhance the inhibitory effect of CK against lung cancer.

## 2. Materials and Methods

### 2.1. Materials and Chemicals

Carboxymethylcellulose sodium (Mw = 90,000), adipic dihydrazide, sodium periodate (NaIO4, Analytical Reagent, AR), N-(3-dimethylaminopropyl)-N-ethylcarbodiimide (EDC) and 1-Hydroxybenzotriazole hydrate (HOBt) were purchased from Macklin Biochemical Co. Ltd. (Shanghai, China). β-Cyclodextrin (Mw = 1135), DMSO (AR) were obtained from Kermel Chemical Reagent Co. Ltd. (Tianjin, China). Cyclohexane (AR), Methanol (AR) were purchased from Fuyu Fine Chemical Co., Ltd. (Tianjin, China) and used as received. Ginsenoside CK (purity > 95%) was provided by the Biomedical Research Institute of Northwest University (HPLC ≥ 95%, Xi’an, China). RPMI1640 medium and penicillin/streptomycin were purchased from GIBCO (Grand Island, NY, USA). Hoechst 33342 was obtained from Beyotime Biotechnology (Shanghai, China). MTT, 4′,6-diamidino-2-phenylindole (DAPI) and Fluorescein isothiocyanate (FITC) were purchased from Beijing Solarbio Science & Technology Co., Ltd. (Beijing, China).

### 2.2. Modification of Carboxymethylcellulose Sodium

The modification of CMC with hydrazine groups was carried out by a previous method [20]. Briefly, 1 g CMC was dissolved in 200 mL of deionized water and 3 g adipic dihydrazide was added. The mixed solution was continuously stirred to make it evenly dispersed at temperature room. Next, HOBt (258 mg suspended in 2 mL of a dimethylsulfoxide: water (1:1) mixture) was added and followed by EDC (262 mg in 2 mL of a dimethylsulfoxide: water (1:1) mixture). If necessary, keep the pH at 6.8 by adding 0.1 M NaOH and stir overnight. After the reaction, the solution was dialyzed with deionized water (8000 MWCO membranes) for 3 days. The final product was obtained by lyophilization. The lyophilized product was subjected to elemental analysis with an elemental analyzer (Vario MACRO cube. Elementar, Langenselbold, Germany) to calculate the degree of modification.

### 2.3. Synthesis of β-CD-CHO

β-CD was oxidized by sodium periodate to obtain β-CD-CHO [21]. Briefly, 7.93 mmol β-CD was dissolved in 200 mL deionized water at 60 °C. 24.03 mmol sodium periodate was added when the solution cooled to room temperature. The reaction was carried out in a water bath at 20 °C and keep magnetically stirred for 1 h in the dark. Then, 5 mL ethylene glycol and 27.03 mmol calcium chloride were added respectively to the mixed solution to stop the reaction. The calcium iodate precipitate formed by reaction with iodate was removed by suction filtration. The resulted mixture was dialyzed against distilled water for 6 h, and finally β-CD-CHO powder obtained by lyophilization.

The aldehyde content of β-CD-CHO was determined by hydroxylamine hydrochloride-potentiometric titration. The aldehyde reacts with hydroxylamine hydrochloride to produce oxime and hydrochloric acid (HCl). The release of HCl was titrated with sodium hydroxide to calculate the aldehyde content. Concretely, 0.1 g β-CD-CHO was added to 25 mL of hydroxylamine hydrochloride solution at 0.25 mol/L. The mixture was stirred at room temperature for 2 h and titrated. The amount of HCl in the mixed solution was measured by potentiometric titration method with standard NaOH solution at 0.05 mol/L. The volume (V, mL) of NaOH solution consumed and the corresponding pH of the solution were recorded, and the titration was stopped when the pH of the solution reached 5.0. The pH-V titration curve was plotted with the volume of the NaOH solution as the abscissa and the pH as the ordinate. Then, the curve was further differentiated, and the peak value of the obtained differential curve was the consumption volume of the NaOH solution. The aldehyde content of β-CD-CHO was calculated by Equation (1).
(1)$$\left[-\mathrm{CHO}\right]=\frac{V_{\mathrm{NaOH}}\times c}{m}$$
where [-CHO] (mmol/g) is the aldehyde content of β-CD-CHO; *V*_NaOH_ (mL) is the peak value of differential curve; *c* (mol/L) is the concentration of NaOH; *m* (g) is the weight of β-CD-CHO. The experiments were done in triplicate.

### 2.4. Preparation of CMC-β-CD Nanogel Loaded with CK

The CMC-β-CD/CK Ngs (CK-Ngs) were prepared via a water-in-oil inverse microemulsion method [22]. Firstly, Span 80 (9.33 mmol) and Tween 80 (0.31 mmol) were dissolved in 18 mL of cyclohexane. A certain amount of CK was dissolved in methanol, then added dropwise to 2 mL β-CD-CHO under probe ultrasound (ultrasound time is 2 min, working 2 s interval 1 s). The resulting mixed solution was added to cyclohexane and sonicated with the probe for 5 min to obtain a microemulsion. CMC-NH_2_ was continuously dripped into the above microemulsion, and then the resulted mixture stirred for 24 h at room temperature. After the stirring, the CK-Ngs were obtained by centrifuging (8000 rpm, 5 min) and washed with ethanol for 3 times to remove the excess surfactant. Finally, the CK-Ngs powders were obtained by freeze-drying. The synthesis process of the blank Ngs (CMC-β-CD Ngs) was as described above.

### 2.5. Drug-Loading and Encapsulation Efficiency

The concentration of CK in the CMC-β-CD Ngs was determined using high performance liquid chromatography (HPLC; Agilent 1260 Infinity) [23]. The lyophilized samples were accurately weighed and dissolved in acetonitrile, sonicated for 30 min, filtered with a 0.45 μm membrane filter, and analysed by HPLC. The Eclipse Plus C18 column (4.60 mm × 250 mm, 5 µm) (Agilent, Santa Clara, CA, USA) was used for separation, and gradient elution chromatography was used with mobile phase of water (A) and acetonitrile (B). The flow rate was 1.5 mL/min, the injection volume was 20 µL and the column temperature was set at 35 °C. The ultraviolet wavelength was set to 203 nm. The linear regression equation of the calibration curve for CK in the test range of 5–100 µg/mL was *y* = 4.4311 *x* + 0.4386, R2 = 0.9992 (*y*: the peak area, *x*: CK concentration, µg/mL). Drug loading (DL) and entrapment efficiency (EE) of CK-Ngs were calculated by Equations (2) and (3).
(2)$$DL\left(\%\right)=\frac{weight\;of\;CK\;in\;microspheres}{weight\;of\;microspheres}\;\times \;100\%$$
(3)$$EE\left(\%\right)=\frac{weight\;of\;CK\;inmicrospheres}{weight\;of\;feeding\;CK}\;\times \;100\%$$

### 2.6. Characterization of Materials

The chemical structure of CMC-NH2 were characterized using ^1^H NMR (Bruker Avance III 400, Bruker, Karlsruhe, Germany). The sample was dissolved in D2O for measurement. The characterization of β-CD-CHO was confirmed by Fourier transform infrared spectroscopy (FTIR, Thermo Scientific, Wilmington, DE, USA), accumulating 36 scans from 400 to 4000 cm^−1^ with a resolution of 4 cm^−1^.The morphological characteristics of CK-Ngs were observed by scanning electron microscope (HITACH S4800, Hitachi, Tokyo, Japan). The particle size (DLS) and zeta potential were determined in a nano particle size and zeta potential analyzer (Zetasizer Nano ZS90, Malvern, UK). The qualitative assay of cellular uptake of CK-Ngs were done using confocal laser scanning microscopy (CLSM, A1R+N-SIM-S, Nikon, Tokyo, Japan).

### 2.7. In Vitro pH-Responsive Drug Release

The in vitro drug release of CK from CMC-β-CD Ngs was assessed by membrane dialysis method [24]. The CK was released in PBS (pH 7.4) and acetate buffer (pH 5.8), containing 1% Tween 80 for more than 120 h. In Brief, 2 mL CK-Ngs suspensions (5 mg/mL) were transferred into dialysis bags (MWCO 3500 Da) and then immersed in 50 mL tubes containing 40 mL release medium. The tubes were placed in an orbital water bath shaker at 37.0 ± 0.5 °C and shaked it at a speed of 120 rpm. Then, 5 mL of release medium was taken out and replaced it with an equal amount of fresh release medium at the time points of 0, 3, 6, 12, 24, 36, 48, 60, 72, 96, 120 and 144 h respectively. The cumulative release percentage of CK from the dialysis bags was determined by HPLC.

### 2.8. Cell Culture

Normal fibroblast cell L929 was selected to test the toxicity of blank Ngs (CMC-β-CD Ngs), and human lung epithelial cancer cell A549 and PC-9 were selected to test the anticancer effect of drug-loaded Ngs (CK-Ngs). L929, A549 and PC-9 cells were cultured in RPMI 1640 supplemented with 10% FBS and 1% penicillin/streptomycin in a 5% CO_2_ atmosphere incubator at 37 °C.

### 2.9. Cytotoxicity Assay

The cytotoxicity of blank Ngs and the inhibitory effect of CK-Ngs on tumor cells were analyzed by MTT method [25]. L929, A549 and PC-9 cells were seeded in 96-well plates at a density of 1 × 10^5^ cells per well according to a certain concentration gradient, and cultured overnight. Subsequently, 100 μL of fresh 1640 medium was added, and blank Ngs of various concentrations were added to L929 cells and incubated for 24 h or 48 h, CK or CK-Ngs were added to A549 and PC-9 cells and incubated for 24 h, 48 h and 96 h. Then, 50 μL MTT (5 mg/mL) solution were added and incubated at 37 °C for 2–4 h. Finally, the supernatant from each well was removed and replaced with 150 μL DMSO. The absorbance(A) was measured at 490 nm using a microplate reader (Power Wave XS2, Bio-Tek Instruments Inc., Winusky, VT, USA).

### 2.10. Apoptosis Test

Hoechst 33342 and AO/EB staining were used to analyze the morphological characteristics of apoptosis of A549 cells treated with CK and CK-Ngs. The A549 cells were cultured overnight in 2 mL of culture medium at a density of 2 × 10^5^ cells/well in 6-well plates. Then, CK and CK-Ngs were added to the cells for 96 h. Control group treated with culture medium. After the supernatant was removed and cells were washed 3 times with PBS, the cells were stained with 10 μg/mL Hoechst 33342 solution for 15 min in the dark or with AOEB staining solution for 10 min, and still washed three times with PBS. The stained cells were observed with a fluorescence microscope (Nikon, Tokyo, Japan).

### 2.11. Cellular Uptake

FITC was used as a fluorescent marker to label nanoparticles, and the cellular uptake of FITC-labeled CK-Ngs was qualitatively analyzed using a confocal laser scanning microscope (CLSM). A549 cells were seeded in a special laser confocal culture dish (NEST, Wuxi, China) at a density of 3 × 10^4^, and incubated at 37 °C. Remove from the incubator at a specific time point, wash with PBS three times, incubate with 4% paraformaldehyde for 15 min, and then wash with PBS. The nuclei were stained with 2 μg/mL DAPI for 15 min and washed with PBS. Finally, the imaging of the cells was observed by confocal laser scanning microscopy (CLSM, Olympus Fluoview FV-1000, Tokyo, Japan).

### 2.12. Human Lung Cancer Xenograft Mouse Model

In order to explore the anti-tumor effect of CK-Ngs in vivo, a nude mouse model of human lung cancer A549 was established. 4-week-old female BALB/Chinese nude mice (16 ± 2 g) were purchased from Hunan SJA Laboratory Animal Co., Ltd., Changsha, China. After the mice were acclimatized under sterile conditions for one week, A549 cells (1 × 10^7^ cells per mouse) were inoculated into the left forelimb fossa of the mice. When the tumor volume reached about 200 mm^3^, the mice were randomly divided into three groups (*n* = 5). Through intraperitoneal injection, the control, free CK and CK-Ngs (20 mg/kg CK equivalents). During the experiment, the body weight and tumor size of the mice were measured every three days, and the tumor volume was calculated as follows: tumor volume (mm^3^) = (length × width ^2^)/2. After 21 days of treatment, the mice were sacrificed humanely by cervical dislocation, the tumors and major organs were obtained by dissection, weighed and imaged. All experiments were conducted in accordance with the People’s Republic of China Animal Ethics Regulations and People’s Republic of China Animal Ethics Guidelines and were approved by the Animal Ethics Committee of Northwest University (NWU-AWC-20201018).

### 2.13. Histopathological Staining

The heart, liver, spleen, lung and kidney of the mouse were taken out and fixed in 10% neutral formalin for 48 h. After dehydration, they were embedded in paraffin according to standard methods to prepare tissue sections with a thickness of 5 μm. Hematoxylin and eosin (H&E) staining: H&E staining was performed according to standard methods [26], and histopathological changes were observed using Nikon TE2000 fluorescence microscope (Nikon, Japan).

### 2.14. Statistical Analysis

All data are expressed in average standard deviation (average standard deviation). The SPSS version 19.0 software (SPSS Inc., Chicago, IL, USA) was used to perform one-way analysis of variance and Student’s t test for the data of more than two groups. *p* < 0.05 was considered statistically significant.

## 3. Results and Discussion

### 3.1. Synthesis and Characterization of CMC-β-CD Ngs

The brief synthesis of CMC-β-CD Ngs is diagrammed in Figure 2.

CMC was modified by introducing amine groups through adipic acid dihydrazide through an EDC-mediated reaction, which was shown in Figure 2A. The modified result was shown in the Figure 3A. Since EDC was water-soluble, it can promote the reaction to occur in an aqueous medium. Moreover, the by-products are also water-soluble, so the final product can be easily purified by simple dialysis or gel filtration [27]. Compared with unmodified CMC, after grafting adipic hydrazide, new spectral peaks were generated at 2.23 ppm (-COCH_2_-), 2.13 ppm (-COCH_2_-) and 1.13 ppm (-CH_2_CH_2_-). The new peaks in these spectra indicate that the CMC was successfully modified and CMC-NH_2_ was obtained. The raw spectral graphs of ^1^H NMR for CMC and CMC-NH_2_ have been added in the Appendix A. In addition, elemental analysis was used to determine the degree of modification. The results showed that the weight percentage of carbon and nitrogen in CMC-NH_2_ were 39.3% and 8.5%, respectively. According to this, the modification degree was estimated to be 5.2%.

It was reported that NaIO_4_ can transfer the 2,3-hydroxyl group of each glucose to the dialdehyde in a stoichiometric way [28], which was shown in Figure 2B. The FTIR spectra of β-CD and β-CD-CHO were exhibited in Figure 3B. As shown in the spectrum in the figure, compared with the unmodified β-CD, the spectrum of β-CD-CHO with higher oxidation degree showed the stretching vibration of the aldehyde group C=O at 1703 cm^−1^. 2828 cm^−1^ and 2776 cm^−1^ were the stretching vibrations of the C-H bond of the aldehyde group, which was a characteristic absorption peak of the aldehyde group. Meanwhile, the aldehyde content calculated by the potentiometric titration of hydroxylamine hydrochloride is 4.51 mmol/g.

It was confirmed by FTIR that CMC-NH_2_ combined with β-CD-CHO to form an amide bond, and a nanogel containing ginsenoside CK was successfully prepared. Figure 3C showed the comparison of infrared spectra of ginsenoside CK, CMC-β-CD Ngs and CK-Ngs respectively. The peaks at approximately 1657 cm^−1^ for the CK-Ngs corresponded to the characteristic peak (C=O) of the CK molecule at 1641 cm^−1^. The peaks of CK-Ngs at 1612 cm^−1^ correspond to the peaks of CMC-β-CD Ngs at 1606 cm^−1^. These results can confirm that CK was successfully packaged.

### 3.2. Drug-Loading and Encapsulation Efficiency

In this study, CK was successfully wrapped in β-CD-CHO mainly through ultrasound probes. By setting different concentrations of β-CD-CHO and CK, CK nanogels with different drug loading levels were obtained, as shown in Table 1. We can see that when the concentration of β-CD-CHO increased from 3% to 10%, the drug loading was also increasing, but when the CK concentration was 10–15 mg/mL, the drug loading gradually decreased. Moreover, when the concentration of β-CD-CHO was 10% and the concentration of CK was 10 mg/mL, the encapsulation efficiency was the highest, at 70.9%. Therefore, this ratio was used for the preparation of subsequent experiments.

### 3.3. Characterization of CK-Ngs

In order to prepare nanogels with controllable size and morphology, water-in-oil reverse microemulsion technology has been widely used [29]. The morphology of nanogels prepared by reverse microemulsion gelation can be controlled by some factors, such as the volume ratio of the water phase to the continuous phase, the amount of surfactants, and the dropping rate [30]. Therefore, this article used this method to prepare spherical nanogels. The following Figure 4A is the particle size distribution curve of CK-Ngs detected by a particle size analyzer, showing a single narrow peak. The particle size and polydispersity index (PDI) were 193.5 ± 3.8 nm and 0.202 ± 0.052, respectively. The results showed that the prepared nanogel microspheres had uniform particle size distribution and good dispersion [31]. As shown in Figure 4B, the zeta potential as an indicator of surface charge mainly affected the stability of nanoparticles in dispersion. The potential of CK-Ngs measured by particle size analyzer is −20.1 ± 0.32 mV. Because of the carboxyl group of carboxymethyl cellulose, CK-Ngs had a negative charge. As observed by scanning electron microscopy, the morphology of CK-Ngs obtained by the water-in-oil reverse microemulsion method is shown in Figure 4C. It is observed from the figure that CK-Ngs was relatively uniform and spherical, and the particle diameter was slightly smaller than the diameter measured from the DLS data, which may be due to the shrinkage of the gel network due to the drying process of the scanning electron microscope [32]. Smaller particle size was more conducive to escape the capture of the reticuloendothelial system, resulting in more accumulation in tumor tissues than in normal tissues [33]. Therefore, CK-Ngs were easily absorbed by cancer cells. As shown in Figure 4D, the CK-Ngs solution was slightly opalescent, while CK was a turbid solution at the same concentration due to its hydrophobicity.

### 3.4. In Vitro pH-Responsive Drug Release

The in vitro drug release of CK from CK-Ngs in two different pH environments was studied within 140 h, as shown in Figure 4E. In the environment of pH 5.8 and 7.4, the release of CK was reflected in the sustained release of 140 h and it can be clearly seen that the release rate of acidic environment is better than that of neutral environment. At pH 5.8, the cumulative release of CK reached 85.5%. At 48 h, the drug release rate increased significantly and lasted to about 96 h, and then was relatively flat. When the pH was 7.4, the cumulative release of CK was 42.8%, which was about half of the drug release at pH 5.8, reflecting the sustained and slow release of the drug. This is because the Schiff base structure formed by the amino group and the aldehyde group in the acidic environment caused the bond to break due to the unstable structure in the acidic environment, which promoted the release of the drug. Compared with normal tissues, tumor tissues showed abnormally high local acidity, so it can effectively inhibit the growth of cancer cells [34].

To determine the release kinetics, the release data was fitted into various kinetic models. Table 2 represents the release kinetics of CK-Ngs at different pH environments. The drug was released following Higuchi’s square-root kinetics. Further, the value of ‘n’ the release exponent of Korsmeyer–Peppas (0.45 ≤ *n* ≤ 0.89) indicates that nanoparticles released the drug by combination of both diffusion of drug through the polymer and dissolution of the polymer [35,36].

### 3.5. Cytotoxicity Assay

Before biological analysis, first perform toxicity analysis on the nanogel carrier at the cellular level [37,38]. As shown in the Figure 5, set different concentration gradients (0, 2.5, 5, 10, 20, 40, 60 μg/mL) to detect the cytotoxicity effect of the empty nanogel carrier on L929, A549 and PC-9 cells within 48 h by the MTT method. The results showed that when the concentration of the nanogel carrier was as high as 60 μg/mL, there was no significant toxicity to the three kinds of cells. Therefore, the concentration of the subsequently selected CK nanogel is not higher than 60 μg/mL.

As shown in Figure 6A,B, CK and CK-Ngs showed obvious dose-dependent and time-dependent effects on A549 cells. Combined with the analysis of the drug release results in 3.4, 48 h and 96 h were the best time for the IC_50_ value of CK-Ngs in A549 cells. The IC_50_ values of CK and CK-Ngs to A549 were shown in Table 3. When the CK in CK-Ngs was released to the maximum at 96 h, its IC_50_ value to A549 cells was 14.98, while the IC_50_ value of CK was 21.03. This indicated that CK-Ngs was more cytotoxic than CK. As shown in Figure 7, CK-Ngs demonstrated better anti-cancer effect on A549 and PC-9 cells (** *p* < 0.01) than CK at 96 h. CK-Ngs can be internalized into cells through receptor-mediated endocytosis, which may lead to a greater concentration of CK in cells. CK was not recognized by P-glycoprotein during this process, which may lead to higher cytotoxicity [39]. As shown in Figure 6D,E, CK-Ngs had the same effect on PC-9 cells.

In order to see more clearly that the toxicity of CK and CK-Ngs to the two cancer cells showed significant differences over time as shown in Figure 6C,F, the cytotoxicity of CK was significantly stronger than that of CK-Ngs before 72 h. However, when the time was extended to 96 h, the survival rates of A549 and PC-9 cells treated with CK-Ngs were 2.5% and 6%, respectively. While the survival rates of A549 and PC-9 cells treated with CK were 11% and 17%, respectively. The delayed effect of CK-Ngs in inhibiting cancer cells was due to the cellular uptake time of nanogels being longer than the time required for free drug release [40]. Therefore, the prepared CK-Ngs showed a significant inhibitory effect in lung cancer cells.

### 3.6. Apoptosis Test

Hoechst 33342 staining and AO/EB staining were used to analyze the apoptosis of A549 and PC-9 cells treated with CK and CK-Ngs to determine whether the cytostatic effect was related to the induction of apoptosis. As shown in the Figure 8, combined with the analysis of the drug release results in Section 3.4, the dyeing analysis was performed when the drug release amount reached the best state at 96 h. The results of Hoechst 33342 staining for A549 and PC-9 showed that the nuclei of the untreated control group showed light blue fluorescence. However, the nuclei treated with CK and CK-Ngs showed a gradual increase in blue fluorescence. It is not difficult to see that the apoptosis of A549 cells was stronger than that of PC-9. Therefore, the apoptotic cells treated with CK-Ngs were more obvious than those in the CK group.

Because AO and EB can specifically bind to live and dead cells, they produced green fluorescence and red fluorescence, respectively. Therefore, after CK and CK-Ngs treated A549 and PC-9 cells, AO/EB staining was used to distinguish live cells from dead cells. As shown in the Figure 9, 96 h was also selected for staining analysis. Regardless of whether it is A549 or PC-9 cells, compared with the control group, the red fluorescence of CK and CK-Ngs gradually increased, and the increase of CK-Ngs was more obvious. At the same time, it is not difficult to see that A549 had a greater degree of apoptosis than PC-9.

### 3.7. Cellular Uptake

The results of the above cell experiments showed that both CK and CK-Ngs had stronger inhibitory effects on A549 than PC-9, so A549 cells were selected as cancer cells in subsequent experiments. Confocal laser scanning microscopy (CLSM) was used to observe the uptake and distribution of FITC-labeled nanogel particles CK-Ngs into cells at different time periods. The cellular uptake distribution was shown in Figure 10, and the difference in the uptake of CK-Ngs labeled by FITC to cells was time-dependent. After the lung cancer cell A549 was incubated with FITC-labeled CK-Ngs for 1 h, FITC-labeled CK-Ngs gradually began to be internalized by the cells. After co-incubating for 3 h, it can be seen that there is a stronger green fluorescence signal around the blue cell nucleus. With the incubation time of lung cancer cell A549 and FITC-labeled CK-Ngs approaching 6 h, the fluorescence intensity around the nucleus gradually increased. Therefore, these results indicated that the nanogel can easily enter the endosome through cell endocytosis, and the absorption of drugs by lung cancer cell A549 was gradually increased over time.

### 3.8. In Vivo Antitumor Effect

In order to explore the anti-tumor effects of ginsenoside CK and CK-Ngs in vivo, an A549 xenograft tumor-bearing mouse model was used for evaluation, as shown in Figure 10. As shown in Figure 11A, the tumor mass size of the three groups of nude mice decreased sequentially, and the effect of the CK-Ngs group was the best. From Figure 11D, it can be clearly seen that the tumor weights of the three groups are gradually decreasing. Among them, the tumor weight of CK-Ngs in the administration group was the smallest. As shown in Figure 11B, at the end of the experiment, the average tumor volumes of the CK group and CK-Ngs group were 392.6 ± 154.2 mm^3^ and 303.8 ± 173.1 mm^3^, compared with the control group (1157.8 ± 165.4 mm^3^), there were significant differences. At the same time, the tumor inhibition rates of ginsenoside CK group and CK-Ngs group on tumor volume were 66.1% and 73.8%, respectively, and the tumor inhibition rates increased sequentially. Obviously, the tumor volume inhibition rate in the CK-Ngs group was the highest. A significant analysis between CK and CK-Ngs was added in Figure 11B,D in the revised manuscript. As shown in Figure 11B,D, CK-Ngs exhibited higher antitumor effect than CK, significantly decreased the tumor volume (* *p* < 0.05) and tumor weight (** *p* < 0.01). Finally, we found that in Figure 11C, neither the control group nor the two administration groups reduced the body weight of the nude mice. This indicated that the toxicity of the two administration groups to nude mice may be small.

### 3.9. Histopathological Analysis

In order to further explore the toxicity of ginsenoside CK and CK-Ngs to the main organs of tumor-bearing nude mice, Histopathological analysis (H&E) staining was used to analyze and observe the effects of ginsenoside CK and CK-Ngs on the main internal organs (heart, liver, spleen, lung, kidney) of tumor-bearing nude mice. The results are shown in Figure 12. The control group, ginsenoside CK group and CK-Ngs group had no obvious pathological changes in the heart, liver, spleen, lung and kidney of tumor-bearing nude mice. The muscle fibers of the ventricle were tightly arranged, clear structure of renal tubules and glomeruli. There was no obvious liver and kidney cell damage or alveolar structure collapse. Therefore, these results indicated that low-dose ginsenoside CK and CK-Ngs had almost no obvious toxicity to xenograft tumor-bearing nude mice.

## 4. Conclusions

In summary, a novel nanogel for the delivery of anti-cancer drugs was successfully constructed by using hydrazide modified CMC and aldehyde modified β-CD. The structure of the nanogel was analyzed by ^1^H NMR and FTIR, and the results showed that the material was successfully modified and the CK was successfully encapsulated by the nanogel. The nanogel had a suitable particle size (193.5 ± 3.8 nm) and zeta potential (−20.1 ± 0.32 mV) as well as a regular spherical shape. In addition, CK-Ngs showed ideal drug release behavior in the acidic environment, in that the cumulative release rate at pH 5.8 in the sustained release within 140 h was 85.5%, while at pH 7.4 it was only 42.8%. In vitro cytological experiments demonstrated that CK-Ngs exhibited stronger cytotoxicity, and the survival rate of A549 cells under CK-Ngs was only 2.98%. Finally, for the in vivo animal tumor model, the inhibitory rates of CK-Ngs on tumor volume were 73.8%, and H&E staining of the main organs of nude mice indicated that CK-Ngs can be safely used to animals. Therefore, this research provided a new idea that can improve the water solubility of CK and achieve controlled drug release to against lung cancer.

## Figures and Tables

**Figure 1 polymers-13-01784-f001:**
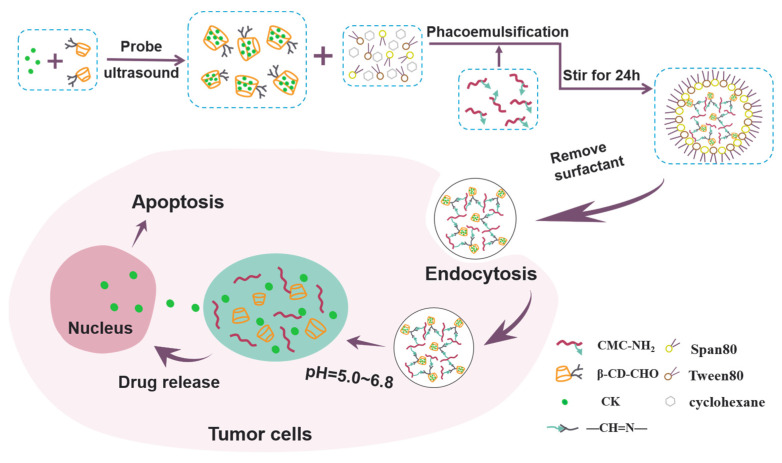
Schematic illustration of preparation of CMC-β-CD Ngs.

**Figure 2 polymers-13-01784-f002:**
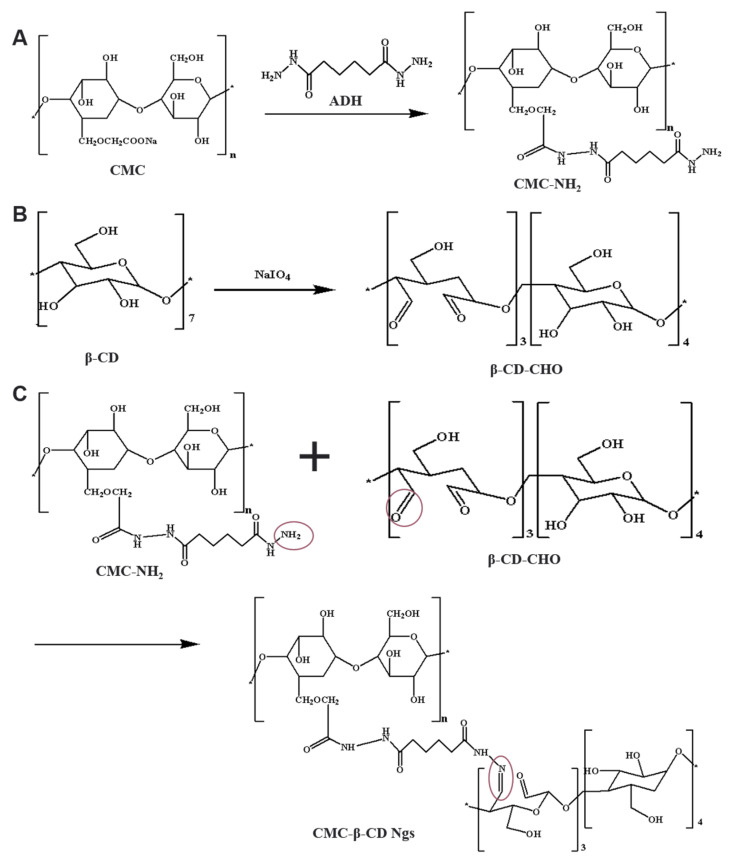
Synthsis of CMC-β-CD Ngs. (**A**) Schematic diagram of adipic acid dihydrazide modification CMC reaction. (**B**) Schematic diagram of the reaction of sodium periodate to oxidize β-CD. (**C**) Schematic diagram of the amide reaction of CMC-NH_2_ and β-CD-CHO.

**Figure 3 polymers-13-01784-f003:**
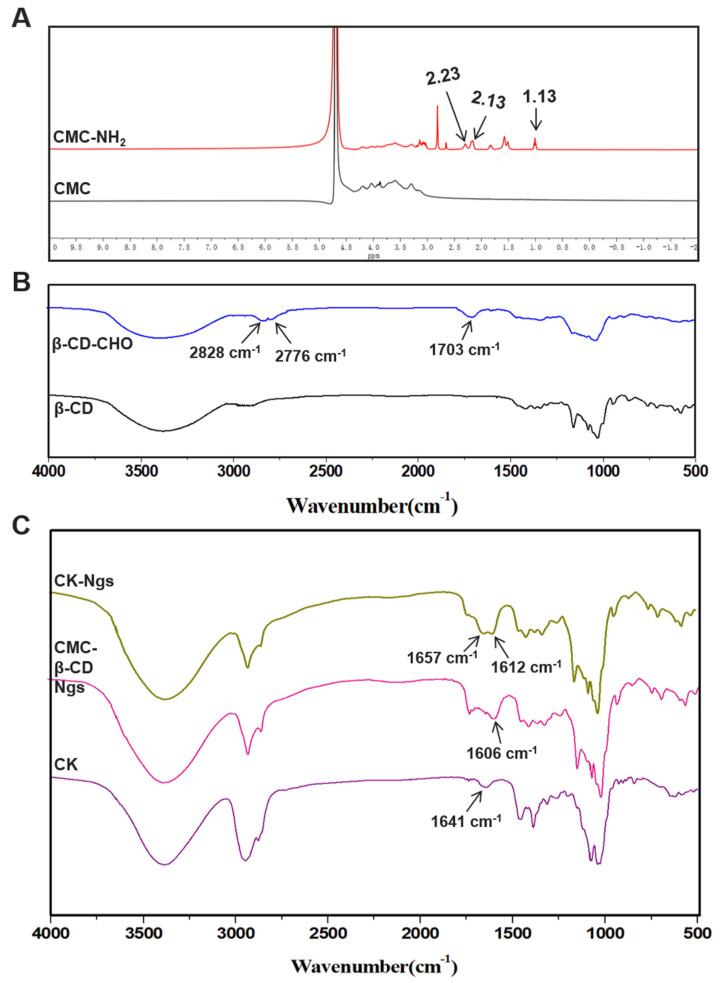
Characterization of modified materials and synthetic materials. (**A**) ^1^H NMR spectra of CMC and CMC-NH_2_. (**B**) Fourier transform infrared (FTIR) spectra of β-CD and β-CD-CHO. (**C**) FTIR spectra of CK, CMC-β-CD Ngs and CK-Ngs.

**Figure 4 polymers-13-01784-f004:**
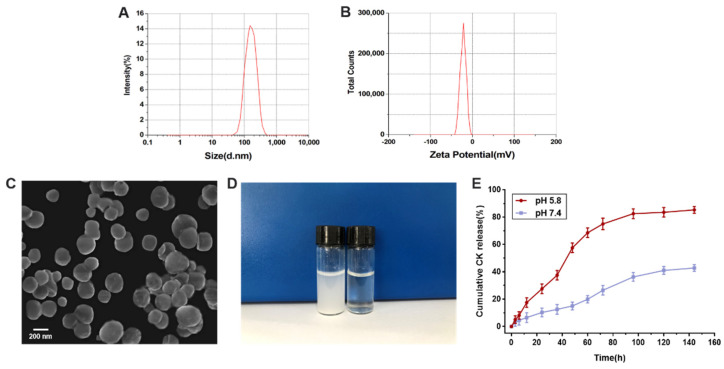
Characterization and drug release of CK-Ngs. (**A**) Size distribution and (**B**) zeta potential distribution determined through DLS and analyzed using Zetasizer. (**C**) Scanning electron microscope (SEM) of CK-Ngs. (**D**) CK and CK-Ngs in water at the same concentration. (**E**) In vitro release profile of ginsenoside CK in CK-Ngs under different pH environments.

**Figure 5 polymers-13-01784-f005:**
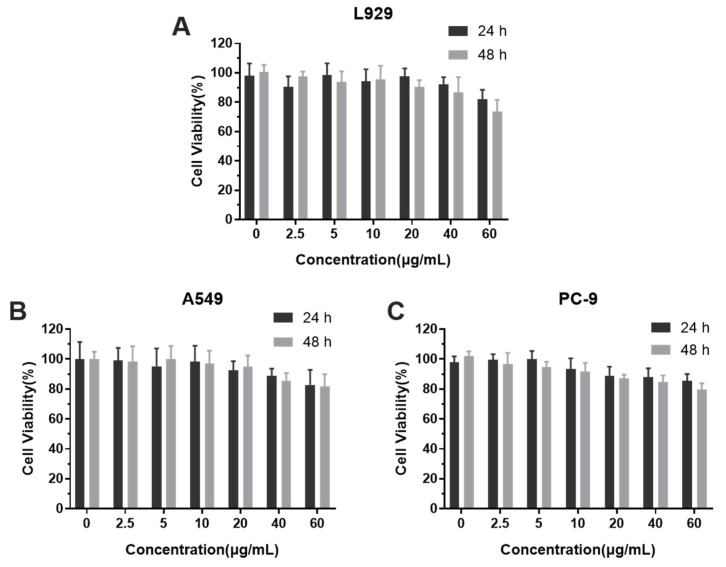
Cytotoxicity of CMC-β-CD Ngs in (**A**) L929 cells, (**B**) A549 cells and (**C**) PC-9 cells at different concentrations after 24 h and 48 h.

**Figure 6 polymers-13-01784-f006:**
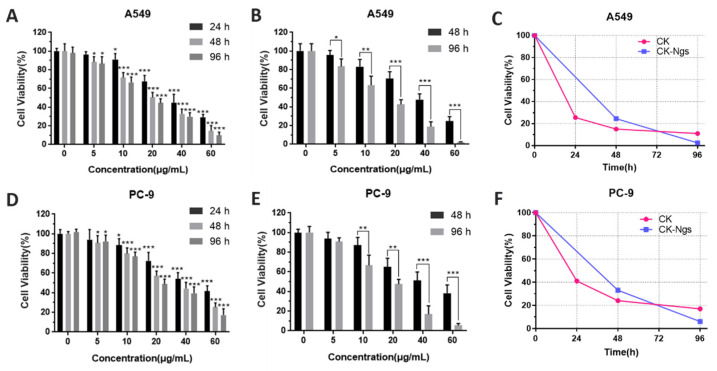
Analysis of the survival rate and line graph of the two drugs on A549 and PC-9 cells. The survival rate of CK in A549 (**A**) and PC-9 cells (**D**). The cytotoxicity of CK-Ngs to A549 (**B**) and PC-9 cells (**E**). The time line chart of the survival rate of CK (**C**) and CK-Ngs (**F**) cells. (* represents *p* < 0.05, ** represents *p* < 0.01, *** represents *p* < 0.001).

**Figure 7 polymers-13-01784-f007:**
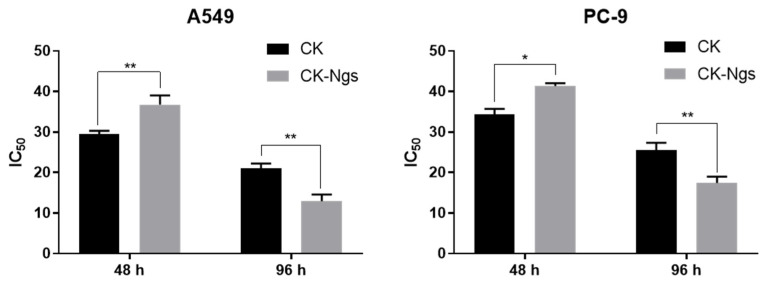
The analysis of IC_50_ value of the two drugs on A549 and PC-9 cells. (* represents *p* < 0.05, ** represents *p* < 0.01).

**Figure 8 polymers-13-01784-f008:**
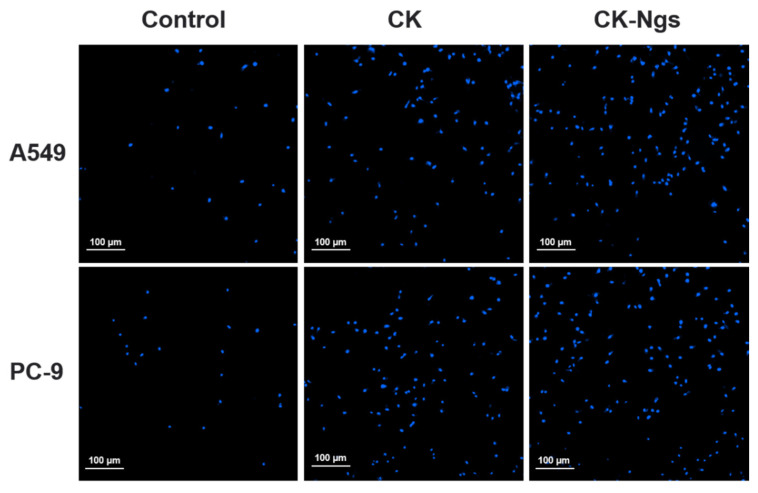
The cell apoptosis induction of CK and CK-Ngs in A549 and PC-9 cells were analyzed by Hoechst 33342 staining (Magnification 20× and scale bar is 100 µm).

**Figure 9 polymers-13-01784-f009:**
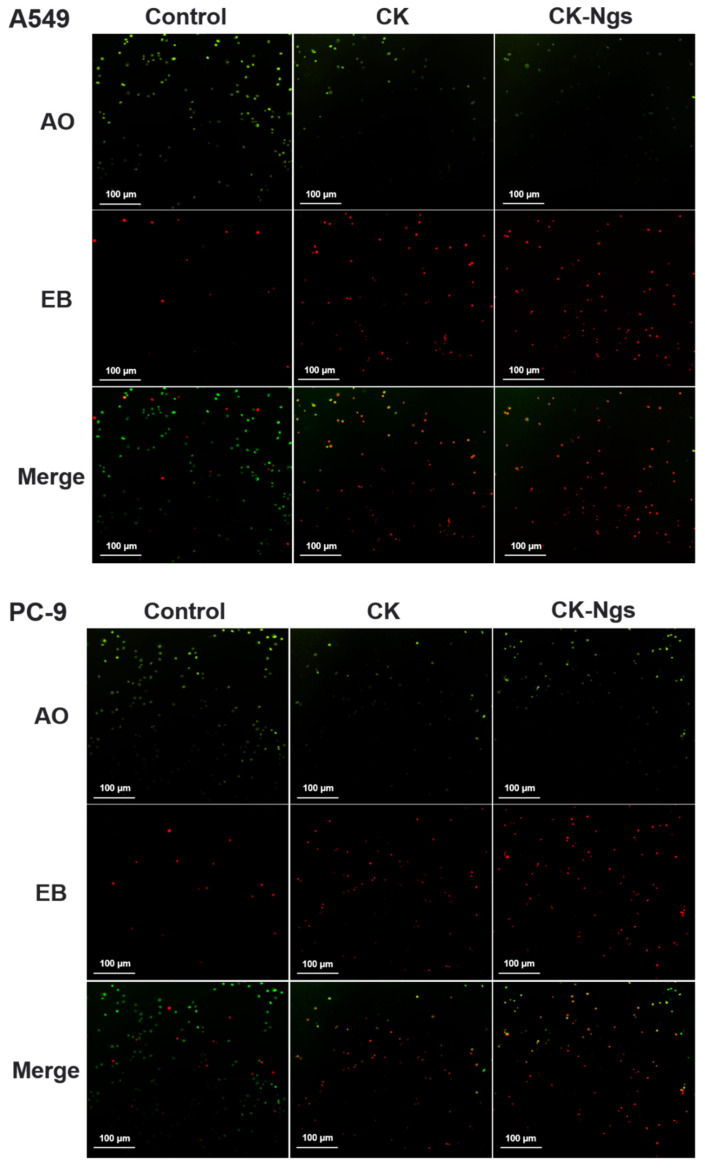
The cell apoptosis induction of CK and CK-Ngs in A549 and PC-9 cells were analyzed by AO/EB staining (Magnification 20× and scale bar is 100 µm).

**Figure 10 polymers-13-01784-f010:**
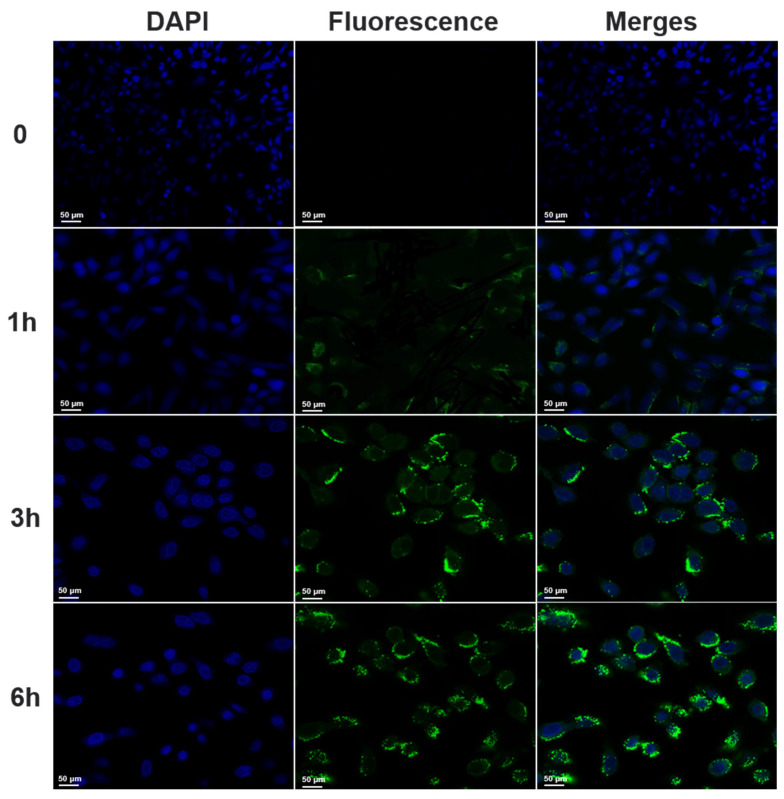
The cellular uptake of CK-Ngs treated with A549 cells for 0, 1 h, 3 h and 6 h were observed by CLSM (Magnification 40× and scale bar is 50 µm).

**Figure 11 polymers-13-01784-f011:**
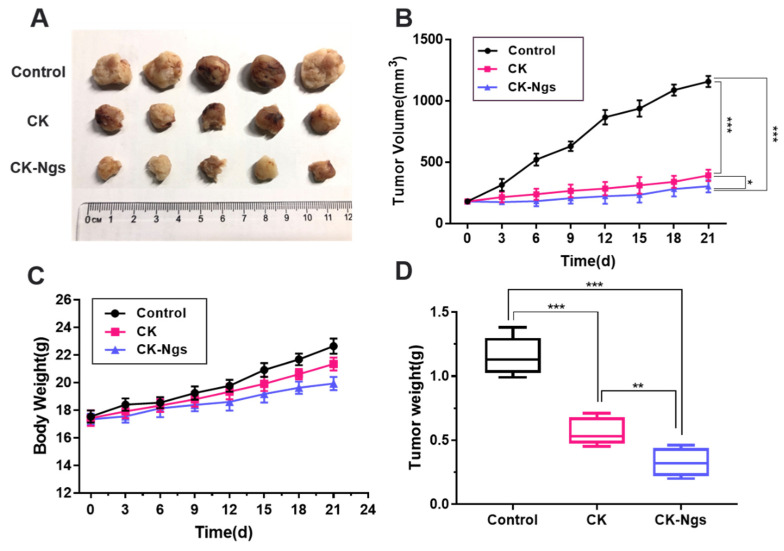
CK and CK-Ngs significantly inhibit the growth of A549 xenografts in vivo. (**A**) The image of A549 tumor-bearing nude mice in different administration groups. (**B**) The effects of different administration groups on tumor volume of nude mice bearing A549 tumor. (**C**) The effect of control, CK and CK-Ngs on body weight of nude mice. (**D**) The effect of different administration groups on tumor weight of A549 tumor-bearing nude mice. (* represents *p* < 0.05, ** represents *p* < 0.01, *** represents *p* < 0.001).

**Figure 12 polymers-13-01784-f012:**
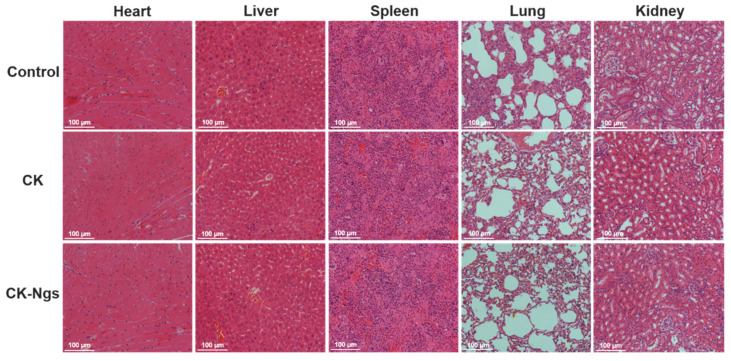
H&E staining of CK and CK-Ngs on main organs of tumor-bearing nude mice (Magnification 20× and scale bar is 100 µm).

**Table 1 polymers-13-01784-t001:** Different concentrations of DA-β-CD and CK drug loading.

	CK	5 mg/mL	8 mg/mL	10 mg/mL	12 mg/mL	15 mg/mL
β-CD-CHO						
3%		4.2%	6%	6.9%	8.5%	6.5%
5%		5.6%	7%	9.8%	10.1%	8.6%
8%		7.1%	9.3%	12.4%	13.8%	11.9%
10%		8.2%	10.5%	16.4%	16.5%	15.8%

**Table 2 polymers-13-01784-t002:** The release kinetics of CK-Ngs under different pH environments.

Release Kinetics					
Formulation Code	Zero Order	First Order	Higuchi’s Square-Root	Korsmeyer Peppas	
	R^2^	R^2^	R^2^	R^2^	*n*
CK-Ngs (pH 5.8)	0.86	0.98	0.99	0.92	0.64
CK-Ngs (pH 7.4)	0.95	0.96	0.99	0.97	0.65

**Table 3 polymers-13-01784-t003:** The IC_50_ of CK and CK-Ngs in A549 and PC-9.

Cell Lines	Incubation Time (h)	IC_50_ (μg/mL)	
		CK	CK-Ngs
A549	24 h	34.64	
	48 h	27.11	39.12
	96 h	21.03	14.98
PC-9	24 h	43.14	
	48 h	34.30	41.35
	96 h	25.56	17.42

## Data Availability

The data that support the findings of this study are available from the corresponding author, upon reasonable request.

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
