# Peer review of "Inhibitory Effect of pH-Responsive Nanogel Encapsulating Ginsenoside CK against Lung Cancer"

_polymers, 2021, doi:10.3390/polym13111784_

Round 1
Reviewer 1 Report
In this work, authors have demonstrated the inhibitory effect of pH-responsive nanogel encapsulated with ginsenoside CK against lung cancer. After careful review, I have found that this study is interesting and well designed and evaluated for the efficacy of ginsenoside CK for lung cancer treatment. This work can be considered for publication in this journal after a minor revision.
1. In addition to Fig. 3A, raw spectral graphs of 1H-NMR for CMC and CMC-NH2 should be provided in Supporting information also.
Author Response
1.In addition to Fig. 3A, raw spectral graphs of 1H-NMR for CMC and CMC-NH2 should be provided in Supporting information also.
The author’s responses: Thanks for your suggestion. The raw spectral graphs of 1H-NMR for CMC and CMC-NH2 have been added in the Supporting Information Fig. S1 as follows.

Reviewer 2 Report
This work is novel and very interesting. In addition, it is in the scope of this journal. Nevertheless, it need some minor corrections before its publication:
1) The authors must to use the journal template.
2) Equations must be defined as equation 1, 2...
3) Why do you use dialysis bags in the drug release tests (section 2.7) and not immersed the systems directly in the medium?
Author Response
- The authors must to use the journal template.
The author’s responses: Thank you. we have revised the manuscript by using the journal template.
- Equations must be defined as equation 1, 2....
The author’s responses: Thanks for your advisement. The equation number have been added in the revised manuscript. Details are as follows.
Drug loading (DL) and entrapment efficiency (EE) of CK- Ngs were calculated by Eq. (2) and Eq. (3).
- Why do you use dialysis bags in the drug release tests (section 2.7) and not immersed the systems directly in the medium?
The author’s responses: Membrane dialysis is commonly used to evaluate the drug release. The relevant references are as below.
[1] Jianmei Zhang, Yijun Wang, Yunyao Jiang; et al. Enhanced cytotoxic and apoptotic potential in hepatic carcinoma cells of chitosan nanoparticles loaded with ginsenoside compound K[J]. Carbohydrate Polymers. 2018, 198: 537-545. DOI: 10.1016/j.carbpol.2018.06.121.
[2] Jiao Guan, Zun-Qiang Zhou, Mao-Hua Chen, et al. Folate-conjugated and pH-responsive polymeric micelles for target-cell-specific anticancer drug delivery[J]. Acta Biomaterialia. 2017, 60: 244-255. DOI: 10.1016/j.actbio.2017.07.018.
In the process of drug release, the intercepted molecular weight of dialysis bag is selected to be larger than the molecular weight of the hydrophobic drug, but less than the molecular weight of the materials, which could prevent the system from flowing into the medium without affecting the drug release. However, when the system is directly immersed in the medium, the pH environment of the medium will change and the subsequent measurement of the drug release value will be unfaithful. Therefore, the drug-loaded nanogel system was put into the dialysis bags for drug release tests.

Reviewer 3 Report
On request of Polymers, I have revised the manuscript titled “Inhibitory effect of pH-responsive nanogel encapsulating ginsenoside CK against lung cancer”, by Ziyang Xue, et al.
General comments.
Ginsenoside CK is a hydrophobic natural occurring molecule, endowed with several anti-cancer activities, but also with poor water-solubility and low bioavailability, which hamper its in vivo application. In this regard, the main scope of this study, was to achieve a water-soluble CK-loaded nanogel carrier, capable to specifically deliver CK to cancer cells.
Following an extensively used approach, CK was easily encapsulated in the hydrophobic cavity of an aldehyde-modified β-cyclodextrin (β-CD-CHO). Interestingly, the actual appealing strategy of the authors, consisted in the idea to prepare the cross-linked architecture of their CK-loaded delivery system, by the formation of Schiff base bonds, between the CK-loaded β-CD-CHO and a hydrazide-modified carboxymethyl cellulose (CMC-NH2). Since the Schiff base bonds are easily hydrolysable at acidic pH, the CK-loaded delivery system would have degraded to the acid pH that characterizes tumor cells, thus realizing a target-specific release of CK.
Actually, the idea to prepare drug delivery systems containing Schiff base bonds, to achieve pH-responsive materials, hydrolysable at acidic pH, and therefore capable to perform a site-specific release of the transported drugs in the cancer cells environments, is not original. However, frequently the hydrolysable bonds are between the drugs and the polymer backbone (Mol. Pharmaceutics 2017, 14, 11, 3866–3878. https://doi.org/10.1021/acs.molpharmaceut.7b00584). Differently in the herein manuscript, such bonds have been included in the cross-linked nanogel and collect the two polymer ingredients. In this case, the drug (CK) is not engaged in covalent bonds with the carrier, but was only encapsulated, thus realizing an improved loading and an easier release, as the in vitro experiments carried out by the authors confirmed (pH 5.8, 85.5 % release after 140 h). The MTT toxicity analysis to evaluate the cytotoxicity against cancer cells, and in vivo animal experiments to investigate the inhibitory effects against tumor were performed.
The fight against cancer is a challenge that continuously engages multitudes of researchers, and the sector is so vast, that the pursuit for developing new synthetic molecules or detecting new natural compounds, able to prevent or treat cancer, should be incessantly enhanced. Hence, there will always be sufficient room for a further experimental article concerning this topic, as well as concerning strategies to improve the water-solubility, the bioavailability, and the target-specific release of bioactive molecules, frequently excessively hydrophobic and with a not favourable hydrophilic/lipophilic balance (HLB). Obviously, only studies particularly well-organized, complete, in terms of performed experiments for evaluating the activity and physicochemical characterization of reported molecules, written using a sufficiently good English, and well-referenced, should be considered for publication. In this regard, the study presents many major and minor issues associated with the lack of mandatory experiments, experimental details, as well as with a misleading discussion of the results, asserting for a non-existing statistical difference between results obtained with pristine CK and CK-Ngls. I am forced to reject the paper, which Polymers should not consider further for publication. A list collecting only some examples of evidenced issues follows.
Abstract. Please, insert some more detail concerning the chemical nature (metabolite of the saponin ginsenoside Rb 1) of ginsenoside CK.
Line 21. Please, use “carriers”, in addition, I suggest using the past tense along the abstract.
Line 23. Please, use “bonds”.
Lines 23 and 24. Please use the forms “hydrazide-modified” and “aldehyde-modified”.
Line 25. The authors should explain in this sentence in which host molecules CK was encapsulated. It is not clear.
Line 28. Please, replace “under” with “in”.
Line 29. Please, add “and” before “the results”.
Line 30. Please, define MTT at the first mention.
Lines 33-34. Please consider the following reformulation of the sentence. “Collectively, the pH-responsive nanogel herein prepared could be considered as a potential nanocarrier for CK to improve its antitumor effects against lung cancer”.
Introduction. Please, reconsider the use of verbs. Frequently, the use the past tense in place of the present, could be a better solution.
Lines 50-51. Badly expressed. Please, reformulate the sentence.
Line 47-51. The most part of concepts contained in these sentences have been repeated starting from line 59. Therefore, I suggest deleting these lines and merging them within the part which starts from line 59.
Lines 51-53. The sentence is non-sense because not finished. In addition, what does “truncated” mean?? Perhaps, the authors meant “truncated cone”?? Please, provide a batter description for β-CD.
Line 55. Please, replace “more and more” with “increasingly”.
Lines 60-61. Please, change the sentence “Such as temperature response, magnetic response and pH response” in “such as temperature, magnetic fields and pH”, and move it within the previous sentence, after “stimuli” and a comma.
Lines 63-65. Badly written. Please, reformulate the sentences, using better the language. For example, “The pH of tumor microenvironment is low (5.8-6.5), due to an augmented glycolysis and hypoxia typical of tumor cells [17, 18]”.
Lines 68-74. As for the abstract, the authors, in the description of the synthetic process followed for achieving the CK-loaded nanogel, must explain in which host molecules CK was encapsulated. It is not clear.
Figure 1. The image is misleading or however not complete. In the images of crosslinked nanogels, the green spheres used to represent CK should be included. Please, remake the Figure 1 accordingly.
Materials and methods. Please, define the abbreviation AR, at the first mention and insert a space before “were” (line 82).
Line 88. Please, define DAPI and FITC at the first mention.
Along the whole experimental part, where possible the authors must insert, the mmol corresponding to the weight of the compound employed. In addition, experiments to determinate the MW of β-CD-CHO and CMC-NH2, should be inserted. Please, perform the requested experiments and insert the description of method adopted.
Line 110-111. “The method of CMC-β-CD/CK Ngs with CK (CK-Ngs) were prepared by a water-in-oil inverse microemulsion method [22]”. In my opinion, there is something wrong in this sentence…
Line 112. Please, replace the comma with a dot.
Line 118. Please, replace “r” with “rpm”.
Line 120. Please, use the past tense.
Line 124. “Sonicated” and not “sonicate”.
Line 125. “filtered” and not “filter”, then, “analysed” and not “detect”.
Section 2.5. Please, improve the description.
Section 2.6. Please, add more details for describing the FTIR and NMR experiments.
Please, following the suggestions provided up now, check the residual experimental procedures and improve their qualities accordingly.
Figure 2 and Figure 3 should be inserted in the Results and Discussion section after their first mention, and not at the end of the Material and Methods Section. More importantly, in panel A and C of Figure 2, the structures of CMC-NH2 are dramatically wrong. Furthermore, the modification shown in panels B and C of Figure 3 are not significant. Curiously, in panel B, while the FTIR spectra appear as in transmittance scale (downward bands), the very small bands related to CHO contribution appear as in absorbance scale (upward bands). A similar phenomenon is observable also in panel C, concerning the contribution of CK in the range 1500-1700 cm-1. In panel A, the integrals of peaks, such as provided by the instruments, must be inserted, to give evidences of what has been commented by the authors.
Line 238. Perhaps the authors meant (CH2OCH2CO-)??
In panel A of Figure 3, there are additional important peaks that authors have not assigned. Please, add such information.
Section 3.4. Regarding the results presented for release studies, to determine the kinetics and mechanisms which govern the release of CK, the most used mathematical models (at least 4/5) should be fitted with the obtained cumulative release (%) curves, and R2 should be computed to determine the model that better fit.
Line 162. Cytotoxicity assay. In which medium was CK dissolved? A549 cells should be treated also with blank Ngs not just with CK or CK-Ngs. Regarding the results of the essay, another cell line was reported (PC-9), which was not indicated in M&M.
IC50 value detected at 48 and 96 h for both cell lines exposed to CK and CK-Ngs, did not evidence statistical differences. Therefore, which are the improvements gained with this study?
Line 193. In the in vivo experiments, what is the vehicle in which the products were dissolved? And its volume? Why did you choose the intraperitoneal injection? The number of the inoculated cells were very high. Could you explain this? And how many days have passed since the start of the treatment? Again, there is not statistical difference between results obtained by administering CK and CK-Ngs.
Even if further issues would be highlighted if I continued, I think that what has been notified so far is more than enough to ask that this work not be considered further for publication on Polymers.
Author Response
1.Abstract. Please, insert some more detail concerning the chemical nature (metabolite of the saponin ginsenoside Rb 1) of ginsenoside CK.
The author’s responses: Thanks. We have provided some details about the chemical nature of CK in the revised manuscript (lines 14-15). Details are as follows.
Ginsenoside CK was one of the intestinal bacterial metabolites of ginsenoside prototype saponin such as ginsenoside Rb1, Rb2, Rc and Rd.
2.Line 21. Please, use “carriers”, in addition, I suggest using the past tense along the abstract.
The author’s responses: Thanks for your suggestion. We have corrected the mistake and used the past tense along the abstract. Details are as follows.
Ginsenoside CK was one of the intestinal bacterial metabolites of ginsenoside prototype saponin such as ginsenoside Rb1, Rb2, Rc and Rd. Poor water solubility and low bioavailability limited its application. The nanogel carriers could specifically deliver hydrophobic drugs to cancer cells. Therefore, in this study, a nanogel was constructed by the formation of Schiff base bonds between hydrazide-modified carboxymethyl cellulose (CMC-NH2) and aldehyde-modified β-cyclodextrin (β-CD-CHO). Water-in-oil reverse microemulsion method was utilized to encapsulate ginsenoside CK via the hydrophobic cavity of β-CD. β-CD-CHO with a unique hydrophobic cavity carried out efficient encapsulation of CK, and the drug loading and encapsulation efficiency were 16.4% and 70.9%, respectively. The drug release of CK-loaded nanogels (CK-Ngs) in vitro was investigated in different pH environments, and the results showed that the cumulative release rate at pH 5.8 was 85.5% after 140 h. The methylthiazolyldiphenyl-tetrazolium bromide (MTT) toxicity analysis indicated that the survival rates of A549 cells in CK-Ngs at 96 h was 2.98% compared to that of CK (11.34%). In vivo animal experiments exhibited that the inhibitory rates of CK-Ngs against tumor volume was 73.8%, which was higher than that of CK (66.1%). Collectively, the pH-responsive nanogel herein prepared could be considered as a potential nanocarrier for CK to improve its antitumor effects against lung cancer.
3.Line 23. Please, use “bonds”.
The author’s responses: Thanks. We have corrected the mistake in the revised manuscript (line 17). Details are as follows.
Therefore, in this study, a nanogel was constructed by the formation of Schiff base bonds…
4.Lines 23 and 24. Please use the forms “hydrazide-modified” and “aldehyde-modified”.
The author’s responses: Thanks. We have used the “hydrazide-modified” and “aldehyde-modified” in the revised manuscript (lines 18-19). Details are as follows.
…between hydrazide-modified carboxymethyl cellulose (CMC-NH2) and aldehyde-modified β-cyclodextrin (β-CD-CHO).
5.Line 25. The authors should explain in this sentence in which host molecules CK was encapsulated. It is not clear.
The author’s responses: Thanks. We have added the relevant information in the revised manuscript (lines 17-20). Details are as follows.
Therefore, in this study, a nanogel was constructed by the formation of Schiff base bonds between hydrazide-modified carboxymethyl cellulose (CMC-NH2) and aldehyde-modified β-cyclodextrin (β-CD-CHO). Water-in-oil reverse microemulsion method was utilized to encapsulate ginsenoside CK via the hydrophobic cavity of β-CD.
6.Line 28. Please, replace “under” with “in”.
The author’s responses: Thanks for your suggestion. We have replaced “under” with “in” in the revised manuscript (line 23). Details are as follows.
The drug release of CK-loaded nanogels (CK-Ngs) in vitro was investigated in different pH environments,…
7.Line 29. Please, add “and” before “the results”.
The author’s responses: Thanks for your advisement. We have added “and” before “the results” in the revised manuscript (line 23). Details are as follows.
…and the results showed that the cumulative release rate at pH 5.8 was 85.5% after 140 h.
8.Line 30. Please, define MTT at the first mention.
The author’s responses: Thanks. Following your suggestion, we have defined MTT in the revised manuscript when it was first mentioned (lines 24-25). Details are as follows.
The methylthiazolyldiphenyl-tetrazolium bromide (MTT) toxicity analysis indicated that…
9.Lines 33-34. Please consider the following reformulation of the sentence. “Collectively, the pH-responsive nanogel herein prepared could be considered as a potential nanocarrier for CK to improve its antitumor effects against lung cancer”.
The author’s responses: Thank you very much. We have rewritten this sentence in the revised manuscript (lines 28-29).
10.Introduction. Please, reconsider the use of verbs. Frequently, the use the past tense in place of the present, could be a better solution.
The author’s responses: Thank you very much. We have used the past tense in the whole introduction in the revised manuscript.
11.Lines 50-51. Badly expressed. Please, reformulate the sentence.
The author’s responses: Thanks. We have reformulated the sentence in the revised manuscript (lines 51-52). Details are as follows.
In addition, proper modification of nanogels could increase the encapsulation efficiency of insoluble anticancer drugs.
12.Line 47-51. The most part of concepts contained in these sentences have been repeated starting from line 59. Therefore, I suggest deleting these lines and merging them within the part which starts from line 59.
The author’s responses: Thanks for your suggestion. We have revised the sentence in the revised manuscript (lines 49-52). Details are as follows.
Nowadays, nanogels had received great attention as an excellent nano-drug delivery system, due to their unique characteristics, including sustaining drug release and adjustable particle size. In addition, proper modification of nanogels could increase the encapsulation efficiency of insoluble anticancer drugs.
13.Lines 51-53. The sentence is non-sense because not finished. In addition, what does “truncated” mean?? Perhaps, the authors meant “truncated cone”?? Please, provide a batter description for β-CD.
The author’s responses: Thanks. We have corrected the mistake in the revised manuscript (lines 42-43). Details are as follows.
β-cyclodextrin (β-CD), which consisted of seven glucose units, possessed a truncated cone structure containing a hydrophobic inner ring and a hydrophilic outer ring.
14.Line 55. Please, replace “more and more” with “increasingly”.
The author’s responses: Thank you for your advisement. We have replaced “more and more” with “increasingly in the revised manuscript (line 45). Details are as follows.
In recent years, increasingly β-CD hydrogels have been developed for drug sustained release.
15.Lines 60-61. Please, change the sentence “Such as temperature response, magnetic response and pH response” in “such as temperature, magnetic fields and pH”, and move it within the previous sentence, after “stimuli” and a comma.
The author’s responses: Thank you. We have revised this sentence in the revised manuscript (lines 53-54). Details are as follows
In order to allow more precise release of drugs around tumors, nanogels that respond to various environment stimuli, such as temperature, magnetic fields and pH had been developed.
16.Lines 63-65. Badly written. Please, reformulate the sentences, using better the language. For example, “The pH of tumor microenvironment is low (5.8-6.5), due to an augmented glycolysis and hypoxia typical of tumor cells [17, 18]”.
The author’s responses: We have revised this sentence according to your suggestion in the revised manuscript (lines 56-57). Details are as follows.
The pH of tumor microenvironment is low (5.8-6.5), due to an augmented glycolysis and hypoxia typical of tumor cells [17,18], therefore, the pH-responsive type is an important feature of the controlled release of the drug at the target.
17.Lines 68-74. As for the abstract, the authors, in the description of the synthetic process followed for achieving the CK-loaded nanogel, must explain in which host molecules CK was encapsulated. It is not clear.
The author’s responses: We have provided the content in the revised manuscript (lines 64-65). Details are as follows.
CK was encapsulated in the hydrophobic cavity of β-CD, which greatly increased the CK loading amount up to 16.4%.
18.Figure 1. The image is misleading or however not complete. In the images of crosslinked nanogels, the green spheres used to represent CK should be included. Please, remake the Figure 1 accordingly.
The author’s responses: Thanks. We have changed the image to make it clearer as below.
19.Materials and methods. Please, define the abbreviation AR, at the first mention and insert a space before “were” (line 82).
The author’s responses: We have given the full name when “AR” was used the first time in the revised manuscript (lines 74-75). Details are as follows.
sodium periodate (NaIO4, Analytical Reagent, AR)
20.Line 88. Please, define DAPI and FITC at the first mention.
The author’s responses:
We have given the full names of all abbreviations when they were used the first time in the revised manuscript (line 83-84). Details are as follows.
MTT, 4',6-diamidino-2-phenylindole (DAPI) and Fluorescein isothiocyanate (FITC) were purchased from…
21.Along the whole experimental part, where possible the authors must insert, the mmol corresponding to the weight of the compound employed. In addition, experiments to determinate the MW of β-CD-CHO and CMC-NH2, should be inserted. Please, perform the requested experiments and insert the description of method adopted.
The author’s responses: Thanks for your suggestions. In the revised manuscript, the molecular weight of the main materials was provided. In addition, the degree of amino modification in CMC-NH2 analyzed by elemental analyzer was about 5.2%. The content of aldehyde group in β-CD-CHO was 4.51mmol/g determined by hydroxylamine hydrochloride potentiometric titration. Since the modification degree of the two materials was low, the modification might not have a great influence on the molecular weight. Therefore, the modified molecular weight was not further measured.
Moreover, the experiments to determine the modification degree of β-CD-CHO and CMC-NH2 were also shown in the revised manuscript and marked the part in red.
22.Line 110-111. “The method of CMC-β-CD/CK Ngs with CK (CK-Ngs) were prepared by a water-in-oil inverse microemulsion method [22]”. In my opinion, there is something wrong in this sentence…
The author’s responses: Thanks for your careful review. We have revised this sentence in the revised manuscript (lines 125-126). Details are as follows.
The CMC-β-CD/CK Ngs (CK-Ngs) were prepared via a water-in-oil inverse microemulsion method.
23.Line 112. Please, replace the comma with a dot.
The author’s responses: We have replaced the comma with a dot in the line 127 of the revised manuscript. Details are as follows.
Firstly, Span 80 (9.33 mmol) and Tween 80 (0.31 mmol) were dissolved in 18 mL of cyclohexane.
24.Line 118. Please, replace “r” with “rpm”.
The author’s responses: Thanks for your advisement. We have replaced “r” with “rpm” in the lines 132-133 of the revised manuscript. Details are as follows.
After the stirring, the CK-Ngs were obtained by centrifuging (8000 rpm, 5 min)…
25.Line 120. Please, use the past tense.
The author’s responses: We have used the past tense in the lines 134-135 of the revised manuscript. Details are as follows.
The synthesis process of the blank Ngs (CMC-β-CD Ngs) was as described above.
26.Line 124. “Sonicated” and not “sonicate”.
The author’s responses: We have corrected the mistake in the revised manuscript (lines 138-139). Details are as follows.
The lyophilized samples were accurately weighed and dissolved in acetonitrile, soni-cated for 30 min…
27.Line 125. “filtered” and not “filter”, then, “analysed” and not “detect”.
The author’s responses: Thanks for your careful review. We have corrected the mistakes in the lines 139-140 of the revised manuscript. Details are as follows.
…, filtered with a 0.45 μm membrane filter, and analysed by HPLC.
28.Section 2.5. Please, improve the description.
The author’s responses: We have rewritten the section in the revised manuscript (lines 137-147). Details are as follows.
The concentration of CK in the CMC-β-CD Ngs was determined using high performance liquid chromatography (HPLC). The lyophilized samples were accurately weighed and dissolved in acetonitrile. After being sonicated for 30 min, the solution was filtered with a 0.45 μm membrane filter, and analysed by HPLC system equipped with an Eclipse Plus C18 column (4.60 mm×250 mm, 5 µm) (Agilent, USA). The mobile phase consisted of HPLC grade water and acetonitrile (50:50% v/v) at a flow rate of 1.5 mL/min. The ultraviolet detection wavelength was set at 203 nm, and the column temperature was maintained at 35 °C. The linear regression equation of the calibration curve for CK in the test range of 5-100 µg/mL was y = 4.4311x + 0.4386, R2=0.9992 (y: the peak area, x: CK concentration, µg/mL). Drug loading (DL) and entrapment efficiency (EE) of CK- Ngs were calculated by Eq. (2) and Eq. (3).
29.Section 2.6. Please, add more details for describing the FTIR and NMR experiments.
The author’s responses: Thanks. We have supplied the details of the FTIR and NMR experiments in the revised manuscript (lines 150-154). Details are as follows.
The chemical structure of CMC-NH2 were characterized using 1H NMR (Bruker Avance III 400). The sample was dissolved in D2O for measurement.
The characterization of β-CD-CHO was confirmed by Fourier transform infrared spectroscopy (FTIR, Thermo Scientific, Wilmington, DE, USA), accumulating 36 scans from 400 to 4000 cm−1with a resolution of 4 cm−1.
30.Please, following the suggestions provided up now, check the residual experimental procedures and improve their qualities accordingly.
The author’s responses: Thanks for your suggestion. We have checked the residual experimental procedures and made some changes in the revised manuscript. Details are as follows.
1.“The tubes were placed in an orbital water bath shaker at 37.0 ± 0.5 °C and shaked it at a speed of 120 rpm.”, which is added in the line 168 of the revised manuscript.
- “…, CK or CK-Ngs were added to A549 and PC-9 cells and incubated for 24 h, 48 h and 96 h.”, which is added in the lines 183-184 of the revised manuscript.
- “Hoechst 33342 and AO/EB staining were used to analyze the morphological characteristics of apoptosis of A549 and PC-9 cells treated with CK and CK-Ngs.”, which is added in the lines 190-191 of the revised manuscript.
31.(1) Figure 2 and Figure 3 should be inserted in the Results and Discussion section after their first mention, and not at the end of the Material and Methods Section.
(2) More importantly, in panel A and C of Figure 2, the structures of CMC-NH2 are dramatically wrong.
(3) Furthermore, the modification shown in panels B and C of Figure 3 are not significant. Curiously, in panel B, while the FTIR spectra appear as in transmittance scale (downward bands), the very small bands related to CHO contribution appear as in absorbance scale (upward bands). A similar phenomenon is observable also in panel C, concerning the contribution of CK in the range 1500-1700 cm-1.
(4) In panel A, the integrals of peaks, such as provided by the instruments, must be inserted, to give evidences of what has been commented by the authors.
The author’s responses: Thanks for your advisements. The followings are my answers to these questions.
(1) Following your suggestion, we have modified the order of the pictures in the manuscript.
(2) We have corrected the mistake in the structures of CMC-NH2. The revised Figure 2A is as follows.
(3) We have re-tested the samples and corrected the mistake. The revised Figure 3B and 3C are as follows. The FTIR spectra of β-CD and β-CD-CHO were exhibited in Fig. 3B. Compared with the unmodified β-CD, the spectrum of β-CD-CHO with higher oxidation degree showed the stretching vibration of the aldehyde group C=O at 1703 cm-1, 2828 cm-1 and 2776 cm-1 were the stretching vibrations of the C-H bond of the aldehyde group, which was a very characteristic absorption peak of the aldehyde group. Meanwhile, the aldehyde content calculated by the potentiometric titration of hydroxylamine hydrochloride is 4.51mmol/g.
It was confirmed by FTIR that CMC-NH2 combined with β-CD-CHO to form an amide bond, and a nanogel containing ginsenoside CK was successfully prepared. Fig. 3C presented the FTIR spectra of ginsenoside CK, CMC-β-CD Ngs and CK-Ngs respectively. The peaks at approximately 1657 cm−1 for the CK-Ngs corresponded to the characteristic peak (C=O) of the CK molecule at 1641 cm−1. The peaks of CK-Ngs at 1612 cm-1 correspond to the peaks of CMC-β-CD Ngs at 1606 cm-1. These results can confirm that CK was successfully packaged.
(4) We have provided the raw spectral graphs of 1H-NMR for CMC and CMC-NH2 in Supporting information. According to the peak area integration, the modification degree of CMC-NH2 is calculated to be about 5.18%, which is consistent with the experimental data (5.2%).
32.Line 238. Perhaps the authors meant (CH2OCH2CO-)??
The author’s responses: We are sorry for the structural mistake of ADH in Fig. 2A and we have corrected it in the revised manuscript. The chemical group with a peak at 1.13ppm is -CH2CH2-. Details are as follows.
33.In panel A of Figure 3, there are additional important peaks that authors have not assigned. Please, add such information.
The author’s responses: Thanks for your careful review. We have marked another important peak in blue and the relevant explanation are as below.
In comparison with the spectrum of unmodified CMC, new peaks at 2.23ppm (-COCH2-), 2.13ppm (-COCH2-), and 1.13 ppm (-CH2CH2-) were derived from grafted adipic hydrazide.
34.Section 3.4. Regarding the results presented for release studies, to determine the kinetics and mechanisms which govern the release of CK, the most used mathematical models (at least 4/5) should be fitted with the obtained cumulative release (%) curves, and R2 should be computed to determine the model that better fit.
The author’s responses: Thanks for your suggestions. To determine the release kinetics, the release data was fitted into various kinetic models. Table 2 represents the release kinetics of CK-Ngs at different pH environments. The drug was released following Higuchi’s square-root kinetics. Further, the value of ‘n’ (the release exponent of korsmeyer-Peppas (0.45 ≤ n ≤ 0.89) ) indicates that nanoparticles released the drug by combination of both diffusion of drug through the polymer and dissolution of the polymer [1-2].
Table 2. The release kinetics of CK-Ngs under different pH environments.
|
Release Kinetics |
|||||||||
|
Formulation Code |
Zero order |
First order |
Higuchi’s square-root |
Korsmeyer Peppas |
|||||
|
|
R2 |
R2 |
R2 |
R2 |
n |
||||
|
CK-Ngs (pH5.8) |
0.86 |
0.98 |
0.99 |
0.92 |
0.64 |
||||
|
CK-Ngs (pH7.4) |
0.95 |
0.96 |
0.99 |
0.97 |
0.65 |
||||
[1] Constantin Mircioiu, Victor Voicu, et al. Mathematical Modeling of Release Kinetics from Supramolecular Drug Delivery Systems[J]. Pharmaceutics, 2019, 11, 140.
[2] J. Siepmann , N.A. Peppas. Modeling of drug release from delivery systems based on hydroxypropyl methylcellulose (HPMC)[J]. Advanced Drug Delivery Reviews, 2001, 48: 139–157
35.Line 162. (1) Cytotoxicity assay. In which medium was CK dissolved?
(2) A549 cells should be treated also with blank Ngs not just with CK or CK-Ngs.
(3) Regarding the results of the essay, another cell line was reported (PC-9), which was not indicated in M&M.
The author’s responses:
Thanks for your suggestions. The corresponding answers are as follows.
(1) CK was dissolved in cell culture medium mixed with 1‰ DMSO. The toxicity of DMSO was negligible.
(2) The detail about the A549 cells treated with blank Ngs was shown in Fig.5B.
(3) We have modified the PC-9 cell lines in section 2.8-2.10 of M&M.
- IC50 value detected at 48 and 96 h for both cell lines exposed to CK and CK-Ngs, did not evidence statistical differences. Therefore, which are the improvements gained with this study?
The author’s responses: We have conducted a significance analysis on the IC50 values between the CK and CK-Ngs groups at 48h and 96h. As shown in Fig. 7, Since the release of CK-Ngs drug takes a certain amount of time, the drug effect of CK-Ngs is not obvious at 48 hours. Whereas, CK completely releases from Ngs at 96h, CK-Ngs demonstrated better anti-cancer effect on A549 and PC-9 cells (**p<0.01) than CK.
37.Line 193. In the in vivo experiments, (1) what is the vehicle in which the products were dissolved? And its volume? (2) Why did you choose the intraperitoneal injection? (3) The number of the inoculated cells were very high. Could you explain this? (4) And how many days have passed since the start of the treatment?
Again, there is not statistical difference between results obtained by administering CK and CK-Ngs.
The author’s responses: The following are my answers to these questions.
(1) CK was dissolved with mixed solution (Ethanol absolute: Polyoxyethylene castor oil: Normal saline 3:6:91) and CK-Ngs were dissolved with PBS (pH7.4). The dose of CK injected to nude mice was 20mg/kg, and CK-Ngs was 122 mg/kg (20 mg/kg CK equivalents). The average body weight of nude mice was 18.5g, and the doses of CK and CK-Ngs were calculated according to the average body weight of nude mice. 0.2mL drug solution was injected into every nude mouse.
(2) Intraperitoneal injection is one of the commonly used methods of drug delivery, through which drugs were absorbed through the peritoneum with a high absorption capacity and can make the medicine reach the tumor site quickly with short rehydration time.
(3) When establishing a nude mouse tumor model, the cell concentration of inoculation is generally 1×106-5×107. The number of inoculated cells used in the manuscript is within the normal concentration range. The references are as follows.
[1] Manling Hu, Jing Yang, Linlin Qu, et al. Ginsenoside Rk1 induces apoptosis and downregulates the expression of PD-L1 by targeting the NF-κB pathway in lung adenocarcinoma[J]. Food&Function. 2020. DOI: 10.1039/c9fo02166c.
[2] Xuqian Deng, Jiaqi Zhao, et al. Ginsenoside Rh4 suppresses aerobic glycolysis and the expression of PD-L1 via targeting AKT in esophageal cancer[J]. Biochemical Pharmacology. 2020. DOI: 10.1016/j.bcp.2020.114038.
(4) When the tumors reached a volume of approximately 200mm3, these mice were injected intraperitoneally with the control, free CK and CK-Ngs every day for three consecutive weeks.
A significant analysis between CK and CK-Ngs was added in Fig.11B and 11D in the revised manuscript. As shown in Fig.11B and 11D, CK-Ngs exhibited higher antitumor effect than CK, significantly decreased the tumor volume (*p<0.05) and tumor weight (**p<0.01).

Round 2
Reviewer 3 Report
Now I have finished the second round revision of this manuscript. I must admit that the authors have worked hard to respond to my comments and have followed almost all of my suggestions. The result is that the work has improved a lot in form, but it is not sufficient, in my opinion, to make this study suitable for publication on Polymers, because it lack of scientific relevance.
Moreover, also in following my suggestions, as that of using the past tense in some parts of the manuscript, the advice was not applied rationally, and the past tense was also used where it did not go well, as in the first sentence of the abstract. Please, correct accordingly.
In addition, the list of bibliographic references does not respect the indications of Polymers. Please, correct accordingly.
Furthermore, As I said in the first round of review, the critical points I pointed out were just examples. This assumed that other criticisms had to be found by the authors and corrected accordingly. Unfortunately, I have only seen corrections where I have notified.
But my biggest perplexity about the publication of this study concerns as abovementioned the scientific relevance of the work itself and its originality. Here I should repeat myself by reiterating what I said in the first revision. If the problems of the manuscript had been only those reported in the list, I would have asked for major revisions and I would not have rejected the work, which I still consider unsuitable for publication on Polymers, but at least on a magazine with a lower impact factor. So, as observable in my report, I have not changed my answers or I have selected "not applicable". This for highlighting that, although I have asked for minor revisions so that authors could correct the residual issues above notified, I go on thinking that Polymers should not consider further this manuscript for publication.